# Peripheral Amino Acid Appearance Is Lower Following Plant Protein Fibre Products, Compared to Whey Protein and Fibre Ingestion, in Healthy Older Adults despite Optimised Amino Acid Profile

**DOI:** 10.3390/nu15010035

**Published:** 2022-12-21

**Authors:** Elena de Marco Castro, Giacomo Valli, Caroline Buffière, Christelle Guillet, Brian Mullen, Jedd Pratt, Katy Horner, Susanne Naumann-Gola, Stephanie Bader-Mittermaier, Matteo Paganini, Giuseppe De Vito, Helen M. Roche, Dominique Dardevet

**Affiliations:** 1UCD Conway Institute and UCD Institute of Food and Health, School of Public Health, Physiotherapy and Sports Science, University College Dublin, D04 V1W8 Dublin, Ireland; 2Neuromuscular Physiology Laboratory, Department of Biomedical Science, University of Padua, 35122 Padova, Italy; 3Institut National de Recherche pour l’Agriculture, l’Alimentation et l’Environnement, Rte de Theix, 63122 Saint-Genès-Champanelle, France; 4Fraunhofer Institute for Process Engineering and Packaging IVV, Giggenhauser Str., 85354 Freising, Germany; 5School of Biological Sciences, The Institute for Global Food Security, Queen’s University Belfast, Belfast BT7 1NN, UK

**Keywords:** aminoacidemia, digestibility, fibre, plant protein, sarcopenia

## Abstract

Plant-based proteins are generally characterised by lower Indispensable Amino Acid (IAA) content, digestibility, and anabolic properties, compared to animal-based proteins. However, they are environmentally friendlier, and wider consumption is advocated. Older adults have higher dietary protein needs to prevent sarcopenia, a disease marked by an accelerated loss of muscle mass and function. Given the lower environmental footprint of plant-based proteins and the importance of optimising dietary protein quality among older adults, this paper aims to assess the net peripheral Amino Acid (AA) appearance after ingestion of three different plant protein and fibre (PPF) products, compared to whey protein with added fibre (WPF), in healthy older adults. In a randomised, single-blind, crossover design, nine healthy men and women aged ≥65 years consumed four test meals balanced in AA according to the FAO reference protein for humans, matched for leucine, to optimally stimulate muscle protein synthesis in older adults. A fasted blood sample was drawn at each visit before consuming the test meal, followed by postprandial arterialise blood sampling every 30 min for 3 h. The test meal was composed of a soup containing either WPF or PPF 1–3. The PPF blends comprised pea proteins with varying additional rice, pumpkin, soy, oat, and/or almond protein. PPF product ingestion resulted in a lower maximal increase of postprandial leucine concentration and the sum of branched-chain AA (BCAA) and IAA concentrations, compared to WPF, with no effect on their incremental area under the curve. Plasma methionine and cysteine, and to a lesser extent threonine, appearance were limited after consuming the PPF products, but not WPF. Despite equal leucine doses, the WPF induced greater postprandial insulin concentrations than the PPF products. In conclusion, the postprandial appearance of AA is highly dependent on the protein source in older adults, despite providing equivalent IAA levels and dietary fibre. Coupled with lower insulin concentrations, this could imply less anabolic potential. Further investigation is required to understand the applicability of plant-based proteins in healthy older adults.

## 1. Introduction

Older adults under-consume both protein and fibre, independent of appetite status [1,2,3,4,5,6]. While it was previously believed that protein ingestion might attenuate appetite, recent studies have shown that neither protein nor fibre reduces food intake in older adults when consumed at recommended daily levels [7,8,9]. It is now well acknowledged that adequate protein intake is key to maintaining a balanced muscle mass and function to prevent sarcopenia due to its role in muscle anabolism (creation) and catabolism (breakdown) [10]. Sarcopenia is characterised by an accelerated loss of muscle mass and strength [11]. It is associated with frailty [12], mobility disability [13], loss of independence [14], reduced quality of life [15], and exacerbated by undernutrition [16]. Albeit many studies have addressed the impact of animal and plant protein on postprandial AA profiles [17,18,19,20,21,22,23,24], to our knowledge, none have examined the potential interactions between protein and dietary fibre on AA availability, which is key for understanding how protein from plant sources, naturally higher in fibre, would appear in circulation. Furthermore, fibre is important in preventing constipation and other chronic diseases common in older populations (e.g., chronic inflammation [25], irritable bowel syndrome [26], obesity, diabetes [27], heart disease [28], and some cancers [29,30] known to challenge negatively muscle mass and function.

Further, while fasted Muscle Protein Synthesis (MPS) rates are similar between healthy young and older adults, the MPS response to protein feeding and exercise [31,32,33,34,35] is often blunted in older individuals (referred to as Anabolic Resistance [36]). In the rested state, healthy older adults require ~0.4 g of protein/kg of body mass to maximise postprandial MPS and overcome anabolic resistance, wherein the anabolic MPS response to protein intake is blunted [37]. In contrast, young adults prompt a similar MPS response after approximately half of that dose [38]. Therefore, older adults require higher protein diets than younger adults to preserve muscle mass and function [39,40]. Moreover, older adults are known to have higher AA splanchnic extraction than their younger counterparts [21,41], resulting in less dietary absorbed AA reaching the peripheral tissues, including muscle tissue, and available for MPS. However, it could be challenging for older adults to consume high amounts of protein due to a decreased appetite [4,5,6] and physiological challenges [42] that are common in ageing. Hence, it is important to optimise the quality of their dietary proteins to constrain the increase in protein consumption as much as possible while ensuring the need for each AA is met [10]. For this, not only protein quantity, but also the quality, are key determinants of AA bioavailability and whole-body protein metabolism [10]. Animal proteins have been shown to more effectively stimulate MPS than plant proteins at the same amount ingested [24,43], which is attributed to plant protein’s suboptimal IAA content [44,45] and digestion and absorption kinetics [18,19,20,46,47]. However, substantially increasing plant protein intake can match such MPS stimulus [22]. Aside from higher protein consumption, other strategies to match plant protein’s ability to stimulate MPS to that of animal protein include improving the AA profile via specific AA fortification or blending different plant proteins [36,48,49].

As the world population increases exponentially [50], with those aged 60+ years estimated to more than double to 2 billion by 2050 [50], animal-based protein may not be available in sufficient quantities to satisfy the growing older population’s dietary requirements. In addition, plant proteins may have less environmental impact [51,52,53] than animal proteins and are more affordable [52,54,55]. Grasso et al. [56] reported that older adults show high acceptability toward ingesting plant protein sources. If the consumer demands [57,58] and dietary recommendations [59,60] favour increasing plant-based foods, plant protein consumption will increase compared to animal-sourced proteins. However, we must ensure the functional effects of plant-derived AA to avoid sarcopenia. There is not enough data to support this transition, as plant proteins’ absorption, digestion, and muscle metabolism in older adults have not been robustly investigated [61].

This study aimed to investigate the arterialise plasma dietary AA profile after ingestion of three different plant protein and fibre (PPF) products compared to whey protein with the same fibre (WPF). The three PPF blends provided different plant AA sources, but matched fibre content. The test meals were matched for leucine content, providing ~2.8 g, the leucine level required to stimulate MPS in healthy older adults [62], and PPF blends were designed to have a high-quality AA profile (as per FAO recommendation [63]) and high in vitro digestibility. Thus, the test meals were designed to address the potential attenuated efficacy of plant proteins on MPS, in comparison to animal proteins, primarily ascribed to the lower leucine content and secondly to unfavourable digestion kinetics compared to animal proteins [24,64]. All protein sources in this study were used as isolates (and, when not possible, concentrates) to reduce differences in digestibility rates [22,24,45,65]. 

## 2. Materials and Methods

### 2.1. Ethical Approval

This study was approved by the University College Dublin (UCD) Sciences and University of Padova (UNIPD) Human Research Ethics Committee and followed the ethical standards laid down in the 1964 Declaration of Helsinki and its later amendments. Participants were informed of the experimental procedures and risks involved in the study. They each gave their written, informed, and voluntary consent before enrolment. This trial was registered at clinicaltrials.gov as NCT05420142.

### 2.2. Participants

Twelve healthy older men and women were recruited to participate in the study through poster and e-mail advertisements. A Consolidation Standards of Reporting Trials (CONSORT) diagram showing the progress from recruitment through the completion of the study is shown in Figure 1. Two participants dropped out during the study due to personal reasons and were therefore excluded from the analysis. A third participant could not finish test meal 3, consuming only 50% of the meal, and was therefore excluded from the analysis. Inclusion criteria were: 65 years of age or older, non-heavy smokers (<10 cigarettes per day) or heavy drinkers (<14 or 21 alcoholic drinks per week in females or males, respectively), and otherwise healthy, according to responses to a standard medical screening questionnaire. Exclusion criteria included self-reported diabetes mellitus, prediabetes, cardiovascular disease, renal disease, gastrointestinal (GI) disease, chronic obstructive pulmonary disease, significant body mass loss in the six months preceding the study, medical condition or use of medication known to impact appetite or energy intake, loss of taste or smell associated with COVID-19, and allergic or unwilling to consume study foods. 

### 2.3. Study Overview 

This was a pilot, randomised, single-blind crossover study conducted in UCD, Ireland, and UNIPD, Italy. Recruitment and the intervention took place between 10 February 2022, and 6 May 2022. Participants completed four identical visits in their respective study centres (Ireland or Italy). The only difference between visits was the protein source, WPF or PPF (1 to 3), added to the test meal. Before the first visit, participants completed a health screening questionnaire, and the protocol was explained in detail. After receiving answers to any questions they had, participants signed the consent form. Body mass was assessed with participants dressed lightly using a calibrated scale (SECA, Hamburg, Germany), and height was measured using a stadiometer (Holtain, Crymych, UK) during the first visit. A schematic overview of the study design is presented in Figure 2. Test days were separated by one to three weeks. Participants presented after an overnight fast (≥10 h) and were immediately cannulated, and a fasted blood sample was drawn before consumption of the test meal. Subsequently, postprandial arterialised blood samples were drawn into Vacutainer Plastic Lithium Heparin plasma collection tubes (4 mL) using the previously inserted cannula every 30 min for 3 h. Samples were centrifuged at 4 °C and stored at −80 °C. The blood arterialisation protocol included a 10 min heating period of the forearm, above the area where the cannula was inserted, before each blood draw. A randomisation tool available at https://ctrandomization.cancer.gov/tool/ was used to randomly assign the test meals to participants over the four visits. Participants were blinded to the identity of the meal containing WPF or PPF 1–3. The study outcome was the appearance of total peripheral AA (IAA and BCAA) concentrations in plasma following the ingestion of a meal containing PPF 1–3 compared to the standard control (WPF).

### 2.4. Test Meal Composition

The test meals were soups of varying protein identities (Table 1). The three PPF products contained a blend of plant proteins (80%) and 20% fibre (40 g of plant protein blend to 10 g of pea fibre). The PPF products differed in the composition of the plant protein (PP) blend (PP1: 67% pea and 33% pumpkin; PP2: 68% pea, 21% oat, and 11% almond; and PP3: 45% pea, 33% soy, and 22% rice). We added 10 g of pea fibre to the whey protein product to match the fibre content of the PPF products. These three PPF products were chosen from six initially developed products due to their higher scores during sensory evaluation by healthy older adults. Fraunhofer Institute provided the PPF products for Process Engineering, and Packaging IVV, Freising, Germany and the rest of the ingredients were bought in local supermarkets, except for the whey protein (Gold Standard 100% Whey, Optimum Nutrition Co., Middlesbrough, UK), which was bought by UCD and distributed evenly between UCD and UNIPD for standardisation. 

All test meals were made with a Knorr low-sodium vegetable stock cube, butter, cornflour, water, and one of three PPF products or WPF (see Appendix A for PPF nutritional composition details). Test meals were between 454 and 498 kcal and contained 30 g of lipids, 15.2 g of CHO, and 30.9 g (WPF-containing test meal), and between 41.1–41.8 g (PPF-containing test meal) of protein, all theoretically containing 3.3 g of leucine. The remaining AA profile of all PPF products was matched as much as possible while optimised for human consumption, as per FAO recommendation [63]. PPF-containing test meals required a higher protein content to achieve the leucine target of 3.3 g (+32% of protein). The cooking methods were as follows: common ingredients were slowly heated in a saucepan until thoroughly combined; then allowed to cool to 40 °C; protein (and fibre for the WPF meal) powders were added, combined, and reheated to no higher than 40 °C immediately before serving. The 40 °C temperature ceiling ensures the structural stability of a protein is maintained without thermal denaturation, which is known to affect protein digestion [66,67].

### 2.5. Analytical Procedures

#### 2.5.1. In Vitro PPF Product Digestibility Assessment 

In vitro digestibility was assessed by an in vitro assay kit (Megazyme, Wicklow, Ireland), described here [68]. Briefly, to simulate the physiological conditions of gastric and intestinal digestion, the samples are first digested at pH 2 by pepsin and then at neutral pH by trypsin and chymotrypsin. Undigested proteins are removed by precipitation with trichloroacetic acid. Amine groups of AA are made available for reaction by the digestion process, which are readily quantified by reaction with ninhydrin to form Ruhemann’s purple. The amount of Ruhemann’s purple formed during this reaction is proportional to the amount of reactive α-AA present in the sample and is quantified by measuring the increase in absorbance at 570 nm. A glycine standard curve is used to determine the linearity of the colourimetric determination of the amines. After correcting the relative reactivity of certain α-AA, an in vitro digestibility score was calculated.

#### 2.5.2. AA Quantification in WPF and PPF 1–3

Three standardised hydrolysation procedures (AOAC, 2000) at 110 °C, as previously described [69], were used to calculate fasting and postprandial AA concentrations in each meal (WPF and PPF 1–3). Briefly, samples were either placed in 6N HCl for 24 h or 48 h (BCAA) or in 6N HCl for 24 h after peroxidation using H_2_O_2_ (methionine and cysteine). After drying and buffer dilution, AA concentrations in each sample were determined by ion exchange chromatography with ninhydrin post-column detection (L-8900 high-speed amino acid analyser, Hitachi).

#### 2.5.3. Blood Analyses

Ultra-performance liquid chromatograph (UPLC) was used to determine plasma AA concentrations before and 30, 60, 90, 120, and 180 min after the test meal was ingested using the UPLC Amino Acid Analysis Solution Waters AccuTag (Manchester, UK) method. First, plasma samples were diluted in TCA (150 µL TCA 50% + 350 µL plasma). The TCA plasma samples were further diluted 2-fold with water and derivatised using Waters AccQ-TagTM ultra reagent kit. Briefly, 10 µL of the sample was mixed with borate buffer containing an internal standard (homoarginine). Then 20 µL of the reconstitute derivatising reagent was added to the mixed solution, followed by a heating step at 55 °C for 10 min. Derivatised AA were analysed with a Waters ultimate 3000 UHPLC system coupled with a fluorescence detector. Briefly, 2 µL of derivatised standards and samples were injected on a column Xbridge C18 (150 × 3.0 mm, 3.5 µm) eluted at a flow rate of 0.98 mL/min. The separation gradient was generated using 4 mobile phases: (A) AccQ-Tag eluant A, (B) water/acetonitrile 90/10 + 2% formic acid, (C) water, and (D) acetonitrile + 2% formic acid for a total run of 23 min. The Waters software, EmpowerTM3, was used for data acquisition and AA chromatogram treatment. AA concentrations were determined based on an AA calibration standard run at 8 concentrations, as follows: 1.25, 2.5, 12.5, 50, 125, 250, 625, and 1250 µM. The proportional molar concentration for each AA was calculated after correction with the internal standard (Homoarginin) and based on the concentration of the standard AA.

Plasma insulin was assessed using a commercial ELISA kit (Mercodia, Sweden), and plasma glucose concentrations were measured by using enzymatic reactions on an autoanalyser (Pentra C400; Horiba). 

#### 2.5.4. Statistical Analysis

All statistical analyses were performed using IBM SPSS (version 27.0.1.0, Dublin, Ireland), and GraphPad Prism 8.2.1 (279, San Diego, CA, USA) was used for graphing. Data were evaluated for the presence of outliers and normality before analysis. Variables not normally distributed according to the Shapiro–Wilk test were log-transformed to achieve normality before the analysis. Variables that were not normally distributed after transformation were analysed using non-parametric tests. AA, glucose, and insulin Incremental Area Under the Curve (iAUC) and AA maximum Concentration (C_max_) were compared between treatments using a one-way repeated-measures Analysis of Variance (ANOVA). In contrast, Time to reach C_max_ (T_max_) was analysed using the non-parametric version of this test (Friedman Test). A two-way repeated-measures ANOVA, with treatment (WPF and PPF 1–3) and time (seven time points) as the within factors, was used to examine plasma AA, glucose, and insulin concentrations over time following the ingestion of the test meal. Where a significant interaction was detected, simple main effects were calculated. All pairwise comparisons were subjected to Bonferroni correction for multiple comparisons. Of note, there was no power analysis performed in this study. Results are presented as means ± SD unless otherwise stated. Significance was set at *p* < 0.05. Further significance was reported as *p* < 0.01 and *p* < 0.001. 

## 3. Results

### 3.1. Participants and PPF In Vitro Digestibility and Composition

The *n* = 9 participants (5 females and 4 males) aged 74 ± 3 years had a BMI of 25.5 ± 3.8. Before the postprandial investigations in the older subjects, in vitro protein digestibility was determined (Table 2). Plant protein blends had higher digestibility than PPF products containing 20% fibre. However, all PPF products scored >100% digestibility (100% is standardised to casein digestibility).

The AA composition of the three PPF blends, compared to the WPF, is presented in Table 3. The PPF products were designed to provide equivalent leucine and highly matched BCAA and IAA, despite being from different plant sources, and align with the optimal values older adults need to promote muscle mass and strength [63]. In further detail, the ΣAA levels provided by the four test meals were similar across the four protein products (47.25–51.19 g). Nevertheless, there were some differences between the AA profiles. The ΣIAA ranged from 19.39–21.32 g, with a higher threonine and isoleucine content in WPF than in the PPF products, and vice versa for methionine and phenylalanine. Therefore, even though plant proteins are known to be deficient in some IAA, such as lysine methionine, this is not the case in this study. Regarding the NIAA (ΣNIAA: 18.97 g in WPF and 21.68–23.68 g in PPF products), higher cysteine concentrations were found in WPF than in the PPF products. The opposite was true for serine, arginine, glycine, glutamic acid, and tyrosine, which were more concentrated in the PPF products than in WPF. As for the BCAA, leucine and valine were evenly distributed in all products (2.82–3.10 and 1.61–1.87 g, respectively), whereas isoleucine was higher in WPF (2.09 g) vs. PPF products (1.46–1.58 g). 

### 3.2. Plasma Leucine, ΣBCAA, ΣIAA, and ΣAA Concentrations

While similar ΣAA levels were provided by the four test meals enriched with the four protein products, the postprandial AA profiles were quite different. The consumption of all meals containing WPF or PPF products increased postprandial plasma ΣAA, ΣIAA, ΣBCAA, and leucine concentrations over the 3 h postprandial period (*p* < 0.001), with differences between meals as indicated by the simple main effect of treatment, where WPF shows a higher concentration at times 60 and 90 min than PPF1 for ΣIAA and ΣBCAA (Figure 3A). ΣIAA and ΣBCAA concentrations for PPF1 were significantly lower in plasma, compared to PPF2 and 3 at 120 min, and overall presented a trend (not statistically significant) towards lower iAUC (Figure 3B). Even though ingested leucine was similar amongst meals, postprandial leucine concentration curves significantly differed between WPF and PPF1 at 60 min, where leucine concentration was higher after WPF ingestion by 74% (Figure 3A). The higher concentration in AA is due to the fast increase in plasma AA post-WPF, followed by a rapid decrease, in comparison to a more sustained elevation of plasma AA post PPF products ingestion (i.e., ~40% slope difference in the change in AA concentration overtime during the first 90 min). Isoleucine, a BCAA, also appeared at higher concentrations postprandially after WPF than PPF ingestion (Figure 4). However, this is likely due to, at least in part, a higher concentration in the source ingested. PPF proteins were more slowly digested than WPF proteins. Overall, the data gathered showed ΣIAA, ΣBCAA, and leucine C_max_ were significantly higher for WPF, and to a lesser extent for PPF3, than for PPF1. Additionally, despite a trend towards later AA appearance after PPF products over WPF, T_max_ values were not significantly different, except for BCAA (WPF > PPF1) (Appendix A). Despite these kinetic differences, iAUC data showed no significant differences between WPF and PPF products or amongst PPF products for ΣAA, ΣIAA, ΣBCAA, or leucine (*p* > 0.05) (Figure 3B). 

### 3.3. Other Amino Acids 

Arginine, tyrosine, and phenylalanine displayed lower postprandial availability following WPF ingestion than PPF products (Figure 4). While arginine’s postprandial concentration curve was significantly lower at all time points after 30 min, tyrosine and phenylalanine postprandial concentration curves had similar C_max_ for all test meals. Yet, after PPF, these concentrations were more sustained, creating a significant difference at the latest time points (150 and 180 min). All three AA were present at higher concentrations in the PPF- vs. the WPF-containing meal (Table 2), which could explain (part of) the observed differences, especially for arginine.

In contrast, postprandial plasma threonine and cysteine concentrations were higher after WPF vs. PPF products consumption (Figure 4). Again, this probably reflects their higher concentration in the WPF-containing meal (Table 2), although it seems that different protein metabolism is occurring with AA from the PPF products. Surprisingly, although methionine was present at higher concentrations in the PPF products, postprandial methionine levels were significantly lower following all PPF products, compared to WPF, as determined by iAUC differences. Notably, the postprandial plasma levels of sulphur AA, methionine, and cysteine decrease past their fasted concentrations after PPF ingestion, resulting in a negative iAUC (Figure 4).

### 3.4. Plasma Glucose and Insulin Concentrations

Overall, there were little differences in postprandial glucose and insulin kinetics in response to the different test meals. However, interestingly, the insulin iAUC is higher after WPF than after PPF1 ingestion (*p* < 0.001). Glycemia curves and iAUC were not significantly different between WPF and PPF product ingestion (*p* > 0.05) and were overall flat, as presented in Figure 5. This is not surprising given the equally and low carbohydrate content in all meals (15.2 g).

### 3.5. Adverse Events and Compliance 

No adverse events were observed during the study in response to the test meal; however, one participant could not finish a test meal due to digestive discomfort.

## 4. Discussion

In the present study, we assessed the net peripheral AA appearance after ingestion of three different PPF products, compared to an animal protein source (i.e., whey proteins with the same pea fibre, WPF) in community-dwelling older adults. Although equivalent amounts of leucine and IAA were ingested, the maximal concentrations of plasma leucine, ΣBCAA, and ΣIAA were significantly lower and delayed following all three PPF products compared to WPF ingestion. Surprisingly, fasting methionine and cysteine plasma levels increased after WPF ingestion; however, these were not elevated after consuming the PPF products. This is contradictory, given that all PPF products contained more methionine than WPF. In addition, the WPF induced higher iAUC insulin levels. Therefore, we report that with plant protein sources, the apparent peripheral bioavailability of some AA was seriously challenged in our older adults, independently (or at least not uniquely dependent) of their dietary intake. 

This study was based on a physiological-driven hypothesis, wherein the meals’ AA profile was designed towards older adults’ specific needs based on their protein requirements to overcome anabolic resistance and efficiently prompt protein synthesis, especially in skeletal muscle. For this, meals were matched for leucine at levels required to optimally stimulate MPS in healthy older adults, ~2.8 g [62], and the remaining AA concentrations were closely similar between meals to avoid the suboptimal IAA profile characteristic of plant-based proteins [45,70,71,72] and optimised as per FAO recommendations [63]. This is unlike previous plant protein bioavailability research in young [23,64,73,74] and older cohorts [24,75,76], in which the net protein content is matched between groups regardless of the IAA profile, with limited research on various plant proteins, besides soy protein, known to broadly differ in their AA composition [77]. Additionally, our study provided the extra proteins with fibre, as older adults often also consume sub-optimal amounts of fibre. A recent study by Pham et al. [17] tested postprandial AA availability in young, healthy men after animal meat proteins (beef and lamb) or a meat analogue (pea protein-based) ingestion. All meals had a highly matched overall protein content (117–118 g) and IAA profile, yet meat ingestion resulted in a significantly higher postprandial AA appearance and higher digestion speed. Thus, plant-based protein intake must be first optimised and then increased to match the postprandial AA response of animal protein, particularly in the elderly, for whom the sensitivity of the anabolic response is triggered and stimulated optimally only with higher amino acidemias. 

The results of the present study add to those previously reported that the digestion speed of whey protein is higher than that of plant-based proteins [18,19,23,48,73,74,76], even after adding fibre. However, as previously noted, others have shown similar findings regarding AA appearance when comparing plant and animal proteins matched for net protein content [17]. One of the factors that could explain the reduced postprandial aminoacidemia concentrations observed after the PPF meals is the greater sequestration of AA in splanchnic tissue [45,78,79], which is known to be higher for plant vs. animal proteins [45,78,79]. Another factor affecting digestion and absorption kinetics is the presence of anti-nutritional compounds common in plant foods (e.g., protease inhibitors or phytic acid) [44,65], which we hypothesise could have remained as residue in the PPF products; however, these were not measured. It is accepted that through the processes by which protein isolates are generated (e.g., extraction, precipitation, several washing steps, temperature treatment, and/or wet fractionation processes), anti-nutritional factors are highly reduced. Nonetheless, protein processing (e.g., dry fractionation processes) could favour anti-nutritional compound accumulation when developing concentrates [80]. Differently processed protein concentrates and isolates were used for PPF product formulation, and the presence of anti-nutritional factors was not assessed directly. Although in vitro digestibility data do not indicate an impact of anti-nutritional factors on digestibility, their effect on in vivo digestibility cannot be ruled out. Other digestive elements that could have influenced the differences in postprandial aminoacidemia concentrations and speed reported here include varying gastric emptying rates, which can be slowed down by higher protein content [81] and/or differences in protein solubility, coagulation, and precipitation (better understood for whey than for plant proteins [82]). We should note that we measured static concentrations of plasma, which cannot determine the dietary protein processes responsible for the AA concentration readouts, for which stable AA isotope-labelled tracing is needed [83]. Nonetheless, postprandial AA appearance and clearance are accepted as a proxy measure for AA digestion and absorption rates [17,84].

Importantly, sulphur AA, methionine, and cysteine were especially adversely affected by the PPF blends. The lack of increase in their postprandial plasma concentrations would classify them as limiting. Methionine’s low postprandial availability was recently flagged after plant-based protein ingestion in young [74] and older adults [17,76], miniature pigs [69], and rainbow trout [85]. Accordingly, we paid particular attention to methionine in the PPF blend design, yet reduced bioavailability of methionine occurred despite this. Thus, increasing methionine intake was still insufficient to improve its bioavailability in our older participants, who have a higher splanchnic extraction of AA, compared to younger subjects [41,86]. Further, ageing is characterised by low-grade inflammation, which could be partly caused by intestinal dysbiosis [87]. Localised intestinal inflammation could generate specific AA demands for immune reactions triggered by a leaky gut [88], particularly glutathione formation [89]. Glutathione contains cysteine, a derivate of methionine; therefore, methionine is converted by transsulfuration into cysteine to sustain its utilisation [90]. Additionally, the ageing population is characterised by increased paracetamol intake [91], which requires glutathione for liver detoxification [92]. Taken together, the overall splanchnic extraction of AA could be enhanced for plant protein sources and associated with the specific sulphur AA requirements during ageing, thus partly explaining our adverse results in increasing methionine and cysteine peripheral availability.

To a lesser degree, threonine is a third IAA that appears to be explicitly affected in this study’s peripheral bioavailability. Its intake with PPF was 30% lower than WPF, but its apparent bioavailability was reduced by almost 60%. For the same reasons cited above for sulphur AA, an increase in splanchnic extraction for this AA can be hypothesised. Indeed, the presence of low-grade intestinal inflammation in older adults, coupled with the presence of plant bioactive harmful to the gut barrier (i.e., lectins [93]), favours the secretion of protective mucins, which are specifically rich in threonine. It has also been shown that in the event of inflammation, a specific need for threonine is initiated [94,95].

Overall, beyond differences in intake, our data indicate that the peripheral AA appearance of threonine, methionine, and cysteine, and to a lesser extent, leucine and isoleucine, are dependent on the protein sources in older adults. Although in this study, leucine concentrations were enough to stimulate MPS in healthy older adults based on previous findings [96,97], the remaining IAA (e.g., methionine and threonine), more so than NIAA [98,99], must also be present in adequate concentrations at the same time to act as precursors and allow efficient protein accretion; otherwise, they are referred to as limiting AA [100,101]. Therefore, a delay in and/or insufficient IAA availability, as seen in this and other studies examining plant protein postprandial curves [18,19,23,48,73,74,76], combined with an ageing effect hindering digestion and absorption [84,102], is expected to have a detrimental impact on the optimal utilisation of all the other AA in the PPF products. However, a recent study by Pinckaers et al. [74] showed that despite limited methionine bioavailability following ingestion of a plant protein source, the muscle anabolic effect was preserved in healthy young males. Whether this is the case with the elderly remains to be studied [103,104].

The insulinogenic effect of leucine is well-characterised [105,106] as an important mediator of the anabolic effect of protein intake on MPS. While the same leucine dose was provided in each meal, the incremental postprandial increase in insulin concentration is higher after the WPF meal than PPF1 ingestion. This may be due to a faster leucine release into the circulation, which stimulates insulin secretion [105,106]. Thus, the insulin-mediated anabolic effect of PPF1 may be sub-optimal.

This study’s strengths include using a cross-over design and standardised meals to reduce intra- and inter-individual variability. The AA profile of WPF and PPF meals was similar and designed to cater to older adults’ specific protein requirements (i.e., higher leucine content), which is a novel aspect of this study, although researched recently in young, healthy people men [17]. Further, a wide variety of protein blends were used, aiming to include those commonly present in households, beyond the scope of those previously researched (e.g., soy and wheat), with the addition of fibre to reflect a real plant food matrix better. Limitations include the lack of a power calculation to assess the number of participants required to validate the conclusions drawn in this manuscript. Further, the presence and composition of anti-nutritional factors were not measured; therefore, their effect in vivo could not be explored. Lastly, despite the addition of fibre, the results presented are not comparable to those of whole foods (e.g., soy, peas, pasta, pumpkin, rice, or oat dishes), which are high in anti-nutritional factors known to greatly limit digestibility and absorption [44,65]. 

## 5. Conclusions

Our results showed that, overall, WPF remained more efficient at increasing postprandial AA concentrations, even when PPF contained 32% more protein (to match the AA profile and reach the leucine target advised for older adults). Further, it is still being determined whether threonine, methionine, and cysteine would increase postprandially at higher plant protein concentrations and whether their suboptimal iAUC has a real effect on MPS. It is evident, however, that older adults require optimal protein and fibre intake for muscle and gut health. Therefore, given the increasing interest in plant-based proteins, both nutrients should be carefully considered when applying strategies to optimise dietary intakes in older adults. Lastly, despite equal leucine doses, the WPF induced greater postprandial insulin concentrations than the PPF products. These data show that adding pea fibre to whey is a potential strategy to increase fibre intake in healthy older adults while mounting an appropriate postprandial aminoacidemic response. However, further research is needed to determine the difference in postprandial aminoacidemia after ingesting whey protein with or without fibre. Lastly, further research assessing the anabolic properties of innovative protein and fibre blends and whole foods in older adults is warranted to understand if shifting towards higher plant protein consumption should be accompanied by higher protein requirements or other novel strategies. 

## Figures and Tables

**Figure 1 nutrients-15-00035-f001:**
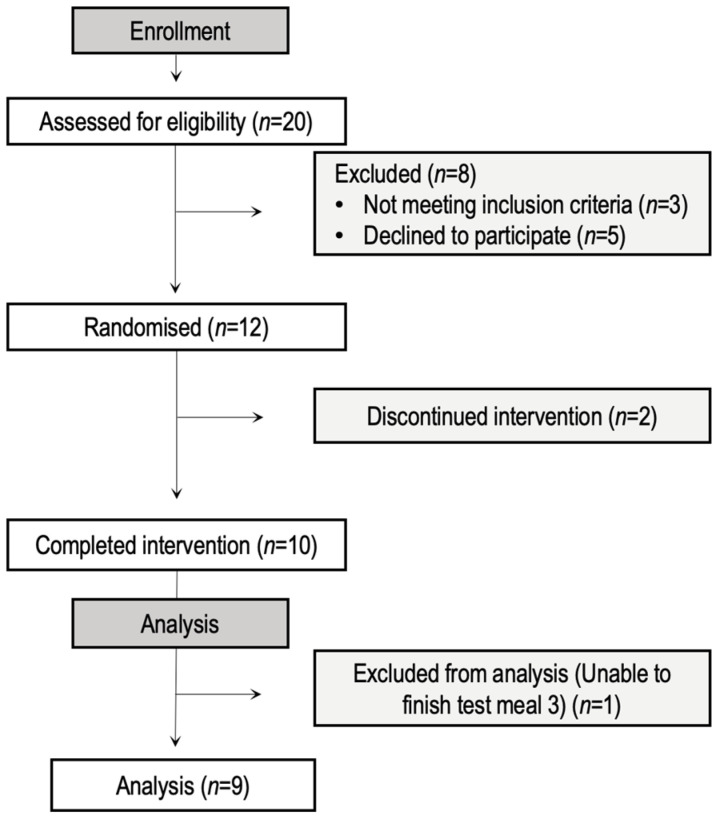
CONSORT diagram.

**Figure 2 nutrients-15-00035-f002:**
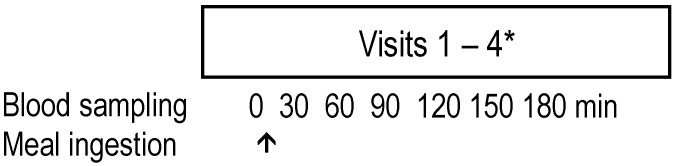
Overview of study design. Each participant underwent four visits (* between one to three weeks apart). The meals were allocated randomly (containing WPF or PPF 1–3). Each PPF product included a blend of plant protein sources and 20% pea fibre. Pea fibre (10 g) was also added to the whey-containing meal. PPF, plant protein fibre; WPF, whey protein fibre.

**Figure 3 nutrients-15-00035-f003:**
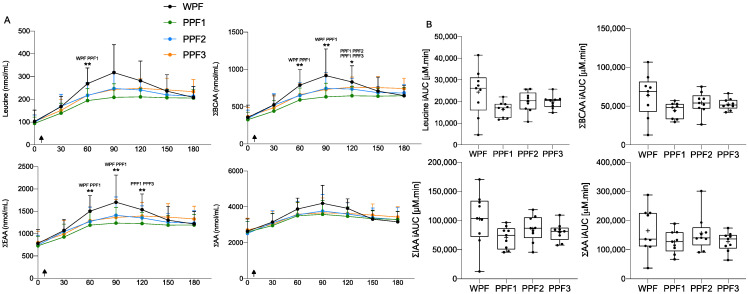
Plasma ΣAA, ΣIAA, ΣBCAA and leucine concentration (**A**) over time (min) and (**B**) their iAUC, in response to a test meal containing WPF or PPF 1, 2, or 3. Values are mean ± SD (*n* = 9). Arrows indicate the ingestion of the meal. Concentration over time data were assessed by 2-way repeated measurements (time × treatment) ANOVA, and iAUC was compared with a 1-way repeated measures (treatment) ANOVA. Significant at *p* < 0.05 (*) or *p* < 0.01 (**). The products’ names above the asterisk indicate the treatments in which postprandial AA concentrations significantly differ at that time. BCAA, branched-chain amino acids; IAA, indispensable amino acids; iAUC, incremental area under the curve; PPF, plant protein fibre; WPF, whey protein fibre.

**Figure 4 nutrients-15-00035-f004:**
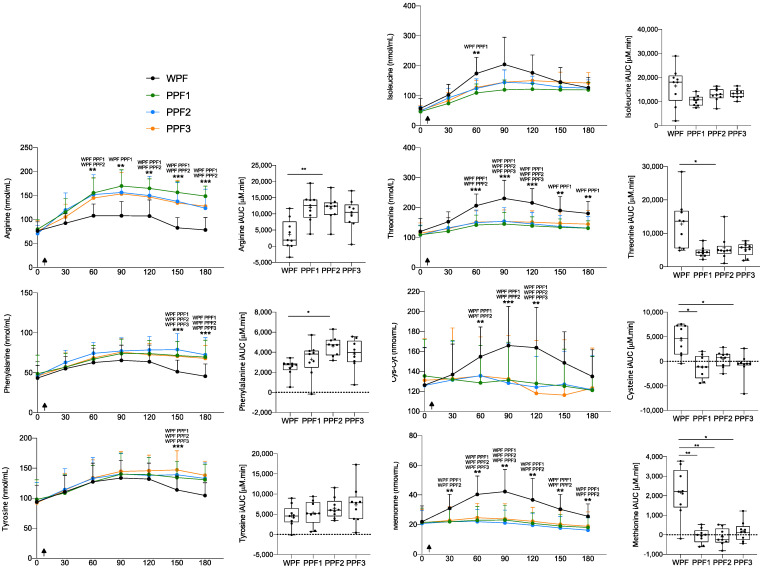
Single AA plasma concentrations, over time (min) and their iAUC, showed a significantly different response post WPF than PPF 1, 2, or 3 meals. Values are mean ± SD (*n* = 9). Arrows indicate the ingestion of the meal. Concentration over time data were assessed by 2-way repeated measurements (time × treatment) ANOVA, and iAUC was compared with a 1-way repeated measures (treatment) ANOVA. Significant at *p* < 0.05 (*), *p* < 0.01 (**) or *p* < 0.001 (***). The products’ names above asterisk indicate the treatments in which postprandial AA concentrations significantly differ at that time point. BCAA, branched-chain amino acids; IAA, indispensable amino acids; iAUC, incremental area under the curve; PPF, plant protein fibre; WPF, whey protein fibre.

**Figure 5 nutrients-15-00035-f005:**
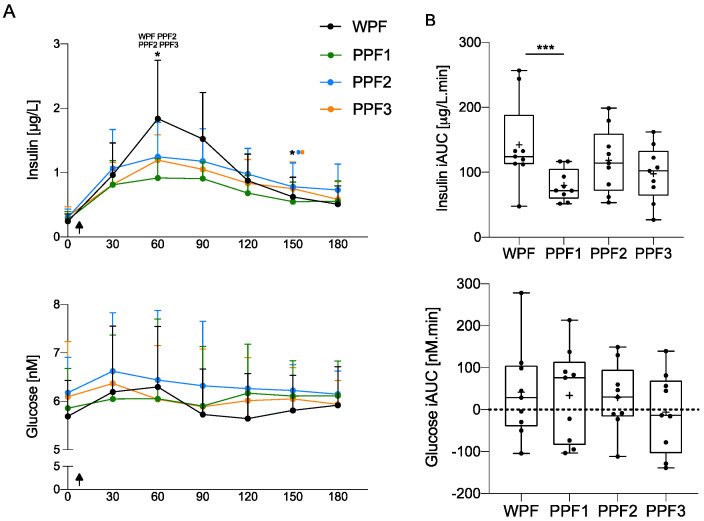
Insulin, but not glucose, plasma concentrations (**A**) over time (min), and (**B**) their iAUC, showed a significantly different response post-WPF than PPF 1, 2, or 3 meals. Values are mean ± SD (*n* = 9). Arrows indicate the ingestion of the meal. Concentration over time data were assessed by 2-way repeated measurements (time × treatment) ANOVA, and iAUC was compared with a 1-way repeated measures (treatment) ANOVA. Significant at *p* < 0.05 (*), *p* or *p* < 0.001 (***). The products’ names above the asterisk indicate the treatments in which postprandial insulin concentrations significantly differ at that time. iAUC, incremental area under the curve; PPF, plant protein fibre; WPF, whey protein fibre.

**Table 1 nutrients-15-00035-t001:** Test meal nutritional compositions ^1^.

	Meal Containing			
Meal Composition	WPF	PPF1	PPF2	PPF3
Lipids, g	30.0	30.0	30.0	30.0
CHO, g	15.2	15.2	15.2	15.2
Protein, g	30.9	41.3	41.8	41.1
Salt + ash, g	0.8	1.4	1.5	1.1
Energy, kcal	454.2	495.7	497.8	495.2

^1^ Macronutrients and energy theoretical values (calculated from the manufacturer’s food nutrition label). CHO, carbohydrate; PPF, plant protein fibre; WPF, whey protein fibre.

**Table 2 nutrients-15-00035-t002:** In vitro protein digestibility of plant protein (PP) and plant protein fibre (PPF) products ^1^.

	In Vitro Digestibility (%)	
	PP	PPF *
PP 1: Pea (67%), Pumpkin (33%)	114 ± 5	110 ± 0
PP 2: Pea (68%), Oat (21%), Almond (11%)	108 ± 3	103 ± 3
PP 3: Pea (45%), Soy (33%), Rice (22%)	109 ± 1	105 ± 0

^1^ All values are means ± SD. * Plant protein fibre products contain 80% plant-based protein and 20% pea fibre. PP, plant protein; PPF, plant protein fibre.

**Table 3 nutrients-15-00035-t003:** Protein products AA composition (quantitatively measured) ^1^.

AA	WPF	PPF1	PPF2	PPF3
BCAA				
Leucine, g	3.00	2.82	3.10	2.88
Isoleucine, g	2.09	1.46	1.58	1.47
Valine, g	1.87	1.61	1.73	1.63
ΣBCAA	6.96	5.89	6.41	5.98
IAA				
Histidine, g	0.58	0.83	0.90	0.85
Threonine, g	2.46	1.52	1.59	1.63
Lysine, g	2.68	2.24	2.40	2.01
Methionine, g	0.65	1.18	1.35	1.43
Phenylalanine, g	1.03	1.84	2.04	1.84
ΣIAA	21.32	19.39	21.1	19.72
NIAA				
Serine, g	1.59	1.90	1.94	1.89
Arginine, g	2.19	3.76	3.30	2.92
Glycine, g	0.58	1.57	1.60	1.44
Aspartic acid, g	3.59	3.86	4.16	3.87
Glutamic acid, g	5.72	6.33	7.40	6.56
Alanine, g	1.64	1.54	1.62	1.62
Proline, g	2.01	1.66	1.97	1.83
Cysteine, g	1.02	0.53	0.49	0.79
Tyrosine, g	0.63	1.10	1.20	0.76
ΣNIAA	18.97	22.25	23.68	21.68

^1^ Values are g per PPF/whey protein added (fresh weight). AA, amino acid; BCAA, branched-chain amino acid; IAA, indispensable amino acid; NIAA, non-indispensable amino acid; PPF, plant protein fibre; WPF, whey protein fibre.

## Data Availability

Data supporting reported results can be provided upon request. Clinical Trial Registration: NCT05420142.

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
