# Peer review of "Peripheral Amino Acid Appearance Is Lower Following Plant Protein Fibre Products, Compared to Whey Protein and Fibre Ingestion, in Healthy Older Adults despite Optimised Amino Acid Profile"

_nutrients, 2022, doi:10.3390/nu15010035_

Round 1

Author Response

Dear Reviewer, 

Please find our answers to all of your points in the attached document. Thank you very much. 

Kind regards, 

Elena 

Reviewer 2 Report

Thank you for the opportunity to review of the article titled "Peripheral Amino Acid Appearance is Lower Following Plant Protein Fibre Products, Compared to Whey Protein and Fibre Ingestion, in Healthy Older Adults Despite Optimised Amino Acid Profile". This study aims to assess the net peripheral Amino Acid appearance after ingestion of three different plant protein fibre products, compared to whey protein with added fibre, in healthy older adults. Protein bioavailability is affected by the digestibility of foods, but the digestibility of proteins in mixed diets, in which multiple foods are combined, is still unknown. The findings represent a useful contribution to the literature.

The following are some comments:

1.      Overall, "Whey" should be corrected as "Whey and fiber".

2.      Abstract: Please briefly state your conclusion. Is it that whey protein has better bioavailability than vegetable protein, even with the addition of fiber, which weakens digestibility?

3.      Table 1: Please show the fiber content.

4.      Table 2: Please show digestibility of Whey in protein (PP) and protein and fiber (PPF).

5.      Table 3: Please show the Amino Acid Scores (i.e., an indices of protein bioavailability based on FAO recommendations) in Whey and PPF 1 to 3, respectively. The authors explain that AA profile of all PPF products was matched as much as possible whilst optimized for human consumption as per FAO recommendation, but there appear to be differences in content in some amino acids (e.g., methionine, phenylalanine). According to the FAO recommendation, it can understand that the balance of essential amino acids in the protein in a food determines the bioavailability of the protein, if digestibility is considered equal. Is it possible that this difference in the proportion of essential amino acids in the protein affected the blood dynamics of the amino acids? Tryptophan is also required to calculate the Amino Acid Score, but is not listed in Table 3.

6.      Study limitations: The authors explain the lack of assessment of MPS as a limitation of this study. However, the study aim is assessing the net peripheral AA appearance after ingestion of three different PPF products, compared to whey protein with added fiber. Thus, it is not study limitation. While, I think the following sentence is a limitation " Differently processed protein concentrates and isolates were used for PPF product formulation, and the presence of anti-nutritional factors was not assessed directly. Although in vitro digestibility data do not indicate an impact of anti-nutritional factors on digestibility, their effect on in vivo digestibility cannot be ruled out." (Page 12, lines 394-397)

7.      Conclusions: Isn’t the conclusion to be led from this study that the addition of dietary fiber to animal protein (whey protein) can provide dietary fiber without compromising the bioavailability of amino acids? To demonstrate that PPF products can be combined with a meal to counteract the reduced bioavailability of IAA, especially methionine and threonine, as the authors say, it would be necessary to test whether the addition of methionine or threonine to plant proteins improves their bioavailability compared to their absence.

Author Response

(The authors gave the same response as above.)
